# *Cichorium intybus* L. Extract Suppresses Experimental Gout by Inhibiting the NF-κB and NLRP3 Signaling Pathways

**DOI:** 10.3390/ijms20194921

**Published:** 2019-10-04

**Authors:** Yu Wang, Zhijian Lin, Bing Zhang, Zhuoxi Jiang, Fanfan Guo, Ting Yang

**Affiliations:** School of Chinese Materia Medica, Beijing University of Chinese Medicine, Beijing 100102, China; wangyuxh@163.com (Y.W.); linzhijian@bucm.edu.cn (Z.L.); 15957717759@163.com (Z.J.); 18434371249@163.com (F.G.); yt12652178@163.com (T.Y.)

**Keywords:** gout, NF-κB, NLRP3, *Cichorium intybus* L., chicory extract, chicoric acid

## Abstract

Background: The production and maturation of interleukin (IL)-1β, regulated by the NF-κB and NLRP3 signaling pathways, lie at the core of gout. This study aimed to evaluate the antigout effect of *Cichorium intybus* L. (also known as chicory) in vivo and in vitro. Methods: A gout animal model was established with monosodium urate (MSU) crystal injections. Rats were orally administered with chicory extract or colchicine. Levels of ankle edema, inflammatory activity, and IL-1β release were observed. Several essential targets of the NF-κB and NLRP3 signaling pathways were detected. Primary macrophages were isolated to verify the antigout mechanism of chicory extract as well as chicoric acid in vitro. Results: Improvements of swelling degree, inflammatory activity, and histopathological lesion in MSU-injected ankles were observed in the treatment with chicory extract. Further, the chicory extract significantly decreased IL-1β release by suppressing the NF-κB and NLRP3 signaling pathways in gout rats. Similar to the in vivo results, IL-1β release was also inhibited by chicory extract and chicoric acid, a specific effective compound in chicory, through the NF-κB and NLRP3 signaling pathways. Conclusion: This study suggests that chicory extract and chicoric acid may be used as promising therapeutic agents against gout by inhibiting the NF-κB and NLRP3 signaling pathways.

## 1. Introduction

Gout, characterized by the deposition of monosodium urate (MSU) crystals within intra- and/or periarticular areas, is an inflammatory disease related to excessive circulating uric acid [1]. Gout flares are not only associated with factors related to reduced quality of life, such as severe pain, limited physical functioning, and a heavy financial burden [2,3], but also contribute to chronic kidney disease, premature mortality, and many metabolic diseases besides hyperuricemia [4,5,6]. According to epidemiological surveys, gout is a serious public health issue and its incidence rate has continued to grow over the past decades [7,8,9].

Presently, fundamental research and clinical treatment of gout emphasize the inhibition of the inflammatory response induced by MSU. Studies suggest that the triggering of interleukin (IL)-1β release, which is activated by MSU stimulation and can further orchestrate a series of inflammatory cascade reactions, lies at the core of gout [10]. The release of IL-1β induced by MSU includes two steps. The first step of nuclear factor kappa-B (NF-κB) activation is required to stimulate the expression of pro-IL-1β [11], and the maturation of IL-1β regulated by the NBD leucine-rich family (NLR) pyrin-containing 3 (NLRP3) inflammasome is the second key step in the initiation of gout flare [12]. Therefore, it is essential to develop medical treatments for gout considering both the NF-κB and NLRP3 signaling pathways.

Nonsteroidal anti-inflammatory drugs (NSAIDs), analgesic drugs, corticosteroids, and colchicine are commonly prescribed to quickly relieve inflammatory pain from gout attacks [13,14]. However, these agents present several serious adverse effects, including renal toxicity and gastrointestinal bleeding [15,16]. Therefore, it is necessary to exploit promising drugs that are safe and effective for gout therapy.

*Cichorium intybus* L., commonly known as chicory, is a perennial herb of the Asteraceae family. The aboveground parts of chicory are generally consumed in Uighur folk medicine as a cholagogic and diuretic agent due to its broad pharmacological action, such as antibacterial, anti-inflammatory, and antioxidant effects, among others [17,18,19]. In addition, chicory is safe to consume as food or medicine [20]. Moreover, the Food and Drug Administration (FDA) has classified chicory extract as “generally regarded as safe” and listed it in the “Everything Added to Food in the United States” category [21]. In previous studies, we found that chicory could significantly decrease serum uric acid, and its anti-hyperuricemia effect may be associated with both the inhibition of urate formation by suppressing xanthine oxidase activity and the promotion of urate excretion by regulating transporter expression [22,23,24,25]. The literature also shows that chicory possesses anti-inflammatory activity [26]. Considering the anti-hyperuricemia and anti-inflammatory effects, as well as the safety of chicory, we propose that chicory could be developed into a promising therapeutic agent for antigout treatment. However, so far, there are no published reports observing the antigout activity of chicory in experimental gout animal models. Chicory extract contains many components, including chicory polysaccharides, chicoric acid, chlorogenic acid, aesculin, and so on. A previous study of ours showed that chicory extract has a high chicoric acid content, which could be an effective constituent for anti-hyperuricemia activity [22]. Like chicory extract, chicoric acid has also been reported to be related to inflammatory inhibition [27,28]. Considering these findings, we hypothesized that chicoric acid may work as an essential active component in chicory extract. Therefore, the present study was designed to observe the antigout effect of chicory extract in experimental rats with gout induced by MSU crystals. On this basis, the effective mechanism of chicory extract in gout through the NF-κB and the NLRP3 signaling pathways could be further explored. At the same time, in vitro cell experiments were carried out to identify if chicoric acid could be the pharmacodynamic material basis of chicory extract in gout treatment.

## 2. Results

### 2.1. Animal Experiments

#### 2.1.1. Effect of Chicory Extract on MSU-Induced Ankle Edema and Inflammatory Activity Index

To assess the extent of edema, the ankle swelling degree (ASD) of the control and treated rats was calculated by measuring ankle circumference. As shown in Figure 1A, the rats given an intra-articular injection of MSU crystals showed a significant increase in ASD, indicating that the development of edema was sustained from 4 to 48 h after MSU injection. It was observed that the chicory extract (15 or 7.5 g/kg) was able to significantly inhibit edema formation at 12, 24, and 48 h after MSU injection. The treatment with colchicine could obviously prevent edema formation at 6 h after MSU injection, which was maintained until the end of the experiment.

Likewise, we also observed the elevation of skin temperature in injected ankles from 4 to 48 h after MSU injection (Figure 1B). The treatment with chicory extract (15 or 7.5 g/kg) and colchicine significantly reduced the skin temperature of injected ankles, observed from 4 to 48 h after MSU injection.

Furthermore, the inflammatory activity index (IAI) of the rats was quantified according to the relevant criteria. As shown in Figure 1C, the IAI in gout rats was significantly increased at 12 and 24 h after MSU injection. However, the treatment with chicory extract (15 or 7.5 g/kg) and colchicine showed a decreasing trend of IAI at 12 h after MSU injection, as well as a significant reduction of IAI at 24 h after MSU injection.

#### 2.1.2. Effect of Chicory Extract on MSU-Induced Synovial Inflammation by Hematoxylin and Eosin (H&E) Staining

Intact ankle architecture with smooth synovium and subsynovial tissues was observed in the control group (Figure 2A). Compared with control group, synovial hyperplasia, edema, and inflammatory cell influx in the synovial lining and subsynovial tissues were found in the ankles of the gout group (Figure 2B). Noticeable relief of inflammatory cell infiltration and synovial hyperplasia was observed in the groups treated with colchicine and a high dose of chicory extract (Figure 2C,D) when compared with the gout group. In addition, a dose–response relationship for chicory extract in tissue reparation was observed, for which there were weaker anti-inflammatory infiltration and antiproliferation abilities in rats treated with a low dose of chicory extract (Figure 2E) than those in rats of the high-dosage chicory treated group. No cartilage/bone destruction was observed in any histological slice.

#### 2.1.3. Effect of Chicory Extract on IL-1β in Gout Rats 

The serum level of IL-1β, an essential pro-inflammatory cytokine in the initiation and progression of gout, was examined by using an ELISA kit to investigate the anti-inflammatory effect of chicory extract on gout. As shown in Figure 3A, the MSU crystals in rats led to a significant elevation in serum IL-1β levels compared with that of the control group. However, the increase of serum IL-1β levels was found to be reduced in rats treated with chicory extract (15 or 7.5 g/kg) and colchicine, which suggests the effect of chicory extract on inhibiting inflammation related to gout.

#### 2.1.4. Effect of Chicory Extract on the NF-κB Signaling Pathway in Gout Rats 

To investigate the underlying mechanisms of the anti-inflammatory effect of chicory on gout, the key targets in the NF-κB signaling pathway were determined by Western blotting. As shown in Figure 3B–D, a noticeable augmentation of protein expression levels of p-IκB and p-p65, as well as increased nuclear translocation of p65, was observed in ankle synovial tissues of the gout group in comparison with the control group. Conversely, compared with the gout group, rats treated with chicory extract showed a significant reduction in the phosphorylation levels of IκB and p65 and the nuclear translocation of p65. Meanwhile, colchicine markedly inhibited the p-p65 protein expression level and nuclear translocation of p65. These findings indicate that regulation of the NF-κB signaling pathway by chicory extract can contribute to its anti-inflammatory effect in gout.

#### 2.1.5. Effect of Chicory on the NLRP3 Signaling Pathway in Gout Rats

We further investigated the mechanisms of the anti-inflammatory effect of chicory on gout from the NLRP3 signaling pathway. The protein expression levels of inflammasome components (ASC, caspase-1, and NLRP3) in ankle synovial tissues of MSU-induced gout rats were measured by immunohistochemistry. As shown in Figure 4A–O, prominent staining of inflammasome components was observed in ankle synovial tissues of rats, which indicated positivity of ASC, caspase-1, and NLRP3 by immunohistochemistry. Figure 4P–R shows that the protein expressions of ASC, caspase-1, and NLRP3 in the gout group significantly increased compared with the control group. The protein expressions of ASC, caspase-1, and NLRP3 significantly decreased in ankle synovial tissues of the groups treated with chicory extract and colchicine. Nevertheless, the group treated with a low dose of chicory extract only showed a pronounced reduction in protein expression of NLRP3.

### 2.2. Cell Experiments

#### 2.2.1. Effects of Chicory Extract and Chicoric Acid on the Proliferation of Macrophages

Macrophages, isolated from the peritoneal cavity of rats, showed good phagocytic function when co-cultured with chicken red blood cells (Figure 5A). Characterization of macrophages was carried out by detecting with immunohistochemistry the antigen CD68, a biomarker on the surface of macrophages. As shown in Figure 5B, prominent staining of CD68 was observed in macrophages. 

As shown in Figure 5C, chicory extract had no significant effect on macrophage proliferation in the concentration range of 200–400 μg/mL. As shown in Figure 5D, chicoric acid significantly inhibited the proliferation of macrophages at a concentration of 400 μM. Therefore, we chose 400 μg/mL and 200 μM as the intervention concentration of chicory extract and chicoric acid, respectively, in the following experiments. 

#### 2.2.2. Effects of Chicory Extract and Chicoric Acid on the NF-κB and the NLRP3 Signaling Pathways in Macrophages

As shown in Figure 6A, IL-1β levels in macrophages stimulated with MSU increased significantly. However, the increase in IL-1β release was obviously reduced in MSU-stimulated macrophages treated with chicoric acid. However, macrophages treated with chicory extract showed a decreased trend of IL-1β release. These findings validate the anti-inflammatory effect of chicory extract in vitro and further demonstrate that chicoric acid could be an effective constituent in chicory extract for antigout treatment.

Furthermore, representative targets in the NF-κB and the NLRP3 signaling pathways were investigated by Western blotting to elucidate the mechanism of the antigout effect of chicory extract and chicoric acid in vitro, respectively. As shown in Figure 6B,C, phosphorylation of p65 and NLRP3 protein expression in macrophages stimulated with MSU was significantly enhanced, whereas treatment with chicory extract and chicoric acid could reduce the obvious elevation of p-p65 and NLRP3 protein expressions in MSU-stimulated macrophages, which indicates the dual inhibition of IL-1β via the NF-κB and NLRP3 signaling pathways. 

## 3. Discussion

Gout is a chronic disease associated with elevated uric acid levels and further MSU deposition in and around the joints, leading to inflammatory cell infiltration and consequent swelling and pain. In the past 50 years, despite the fact that gout is a public health problem worldwide, few new specific medical treatments have been found for the clinical management of gout besides colchicine. However, colchicine usually leads to serious adverse effects when applied for gout treatment [29,30,31,32]. Moreover, approximately 50% of gout patients may be non-compliant with the prescribed medication when gout flares recur [33]. Under these circumstances, many gout patients turn to folk medicine, which is mainly derived from natural plants, for help.

*C. intybus* L., commonly known as chicory, is often used as a cholagogic and diuretic agent, which conforms to the therapeutic principle of hyperuricemia and gout in Chinese traditional medicine. Our previous studies demonstrated the promising effect of chicory on lowering serum urate levels [22,23,24,25], and other reports have also shown the anti-inflammatory effect of chicory extract [26]. Combining the traditional use of and modern pharmacological research on chicory, we posited that chicory may be a safe antigout agent that can address both the symptoms and root causes of gout. On the basis of previous research on anti-hyperuricemia, the present study focused on the antigout effect of chicory.

We first analyzed the anti-edematogenic effect of chicory extract in experimental gout animals induced with an intra-articular injection of MSU. Symptoms of the injected ankles, such as ankle swelling degree, skin temperature, and inflammatory activity index, were measured. As expected, the degree of ankle swelling, skin temperature, and inflammatory activity index in gout rats treated with chicory extract showed a significant reduction, which indicates the possible antigout effect of chicory extract. Further observations were made by H&E staining. Decreased inflammatory cell infiltration and relieved synovial hyperplasia were observed in gout rats treated with chicory extract, which supports the presumed antigout effect of chicory extract.

In addition to edema relief of joints, gout treatment also must inhibit the inflammatory response induced by MSU. MSU stimulation may cause an innate immune response with the release of several pro-inflammatory cytokines and further amplify inflammatory response by producing leukocyte infiltration, leading to tissue damage [34,35]. Among these pro-inflammatory cytokines, IL-1β plays a crucial role in gouty inflammation. For instance, reduced inflammation was found in murine models treated with inhibitors IL-1β [10], and rapid clinical response to IL-1β inhibition was also observed in gout patients [36]. In our study, the chicory extract significantly decreased IL-1β release induced by MSU injection, which verifies the antigout effect of chicory extract. However, we did not observe a statistically significant dose-dependent response of chicory extract in ankle edema and IL-1β release.

As mentioned above, IL-1β lies at the core of gouty inflammation, and inhibition of IL-1β release is an effective therapeutic method for gout treatment. This cytokine is produced as an inactive promolecule, referred to as pro-IL-1β, and is then cleaved into the active form of IL-1β to be secreted. NF-κB is a dimeric transcription factor which can be activated by MSU to increase gene expression of pro-IL-1β [11]. Cleavage of pro-IL-1β is by caspase-1, which is an important component of the NLRP3 inflammasome. This inflammasome is formed through homotypic interactions between the CARD and PYD domains of NLRP3, pyrin, and the adaptor ASC. Reports have shown that the NLRP3 inflammasome can be activated by MSU and can further convert pro-caspase-1 into active caspase-1 [37,38]. Therefore, we further examined IL-1β release in gout via the NF-κB and the NLRP3 signaling pathways, respectively, to explore the mechanism of the antigout effect of chicory extract. The results demonstrate that the activated NF-κB signaling pathway in MSU-injected rats featured elevated phosphorylation levels of IκB and p65, as well as increased nuclear translocation of p65. Further, increased protein levels of ASC, caspase-1, and NLRP3, which represent the activation of the NLRP3 signaling pathway, were also observed in MSU-injected rats. However, these activated pathways could be obviously inhibited with chicory extract treatment. On this basis, it indicates that the antigout effect of chicory extract can be achieved by inhibiting activation of both the NF-κB and NLRP3 signaling pathways in MSU stimulation. Subsequently, cell experiments were performed to verify the antigout mechanism of chicory extract and chicoric acid in vitro. Leucocytes, including neutrophils and macrophages, are the main cell types for the study of gouty inflammation. Due to serious cell death, as well as the release of lysosomal and cytoplasmic enzymes in neutrophils stimulated with MSU crystals, macrophages would be more suitable for the study of gouty inflammation with the principal secretion of IL-1β [39,40]. Primary macrophages were isolated from the rat peritoneal cavity and identified with CD68, a biomarker on the surface of macrophages. Increased release of IL-1β, as well as activation of the NF-κB and NLRP3 signaling pathways, was observed in isolated macrophages incubated with MSU. Similar to the results in vivo, chicory extract markedly reduced protein expressions of essential targets in the NF-κB and NLRP3 signaling pathways at the concentration of 400 μg/mL. The non-significant decrease of IL-1β release in macrophages incubated with chicory extract may have been related to drug concentration. Chicoric acid is a major active ingredient of chicory that has anti-inflammatory properties [28]. The results of the present study show that protein expressions of p-p65 and NLRP3 in MSU-stimulated macrophages were reduced after treatment with chicoric acid. Production of the mature forms of IL-1β also decreased with chicoric acid treatment, which indicates that chicoric acid may be responsible for the antigout effects of chicory extract. Taken together, our findings in vitro provide evidence for the antigout effect of chicory extract and indicate that chicoric acid primarily works via the inhibition of IL-1β release through both the NF-κB and NLRP3 signaling pathways.

## 4. Materials and Methods 

### 4.1. Animal Experiments

#### 4.1.1. Chicory Extract Preparation

The aboveground parts of chicory used in this study were purchased from Jilin, China. The plant materials were authenticated taxonomically by Professor Yong-Hong Yan (Traditional Chinese Medicine Appraisal Teaching and Research Section of Beijing University of Chinese Medicine). A dried chicory plant (1 kg) was ground into powder and extracted with water (10 L) by heating to reflux for 1 h twice. Then, the decoction was filtered and concentrated under reduced pressure [22]. 

#### 4.1.2. Induction of Gouty Arthritis with MSU Crystals and Drug Treatment

The gout model was induced by intra-articular injections of MSU according to Coderre and Wall [41]. Briefly, 100 μL of MSU suspension (25 mg/mL) was injected into the tibiotarsal joint (ankle) of rats. Rats in the control group were injected with equal volumes of vehicle.

Sprague–Dawley male rats (200 ± 10 g) were purchased from Beijing SPF Laboratory Animal Technology Co., Ltd. (Certificate of Quality: SCXK-2016-0002). All rats were housed in a temperature-controlled room at 22 ± 2 °C, with 55 ± 10% relative humidity and a 12 h light–dark cycle, and were supplied with water and food ad libitum for 6 days. All efforts were made to minimize animal suffering. All animal procedures were approved by the Animal Care and Ethics Committee of Beijing University of Chinese Medicine (BUCM–4–2018090301–3032, September 3 2018).

After acclimation, animals were divided into the following five groups, which consisted of 10 rats each: (A) Control group with, (B) gout group with MSU injection, (C) colchicine group with MSU injection and intragastric administration of colchicine (0.8 mg/kg body weight), (D) high-dosage chicory treated group (CH group) with MSU injection and intragastric administration of chicory extract (15 g/kg body weight), and (E) low-dosage chicory treated group (CL group) with MSU injection and intragastric administration of chicory extract (7.5 g/kg body weight). The rats were administered chicory extract or the positive control drug (colchicine) 5 days before the MSU crystal injection and then once daily for 2 days. During the experiment, the control and MSU groups were intragastrically administered an equal volume of purified water.

#### 4.1.3. Assessment of Ankle Edema

Ankle edema formation was assessed as an increase in ankle circumference. We measured the ankle circumference at 0.5 mm below the ankle joint before and 4, 6, 12, 24, and 48 h after MSU injection. Each measurement was repeated three times to obtain an average. ASD was calculated using Equation (1):Ankle swelling degree (%) = [Ankle circumference after injection (mm) − Ankle circumference before injection (mm)] / [Ankle circumference before injection (mm)] ×100%.(1)

#### 4.1.4. Assessment of Inflammatory Activity

Inflammatory activity was assessed by scores obtained 12 and 24 h after MSU injection. The assessment was carried out by three testers separately, according to the following criteria: 0 points: An ankle with visible osseous landmarks; 3 points: An ankle with slight swelling and visible osseous landmarks; 4 points: An ankle with serious swelling and invisible osseous landmarks; 6 points: There was limb edema besides the ankle. The skin temperature was also measured by infrared thermometer to dynamically observe inflammatory activity during the experimental period.

#### 4.1.5. Histopathological Analysis

Rats were sacrificed at 48 h after MSU injection. The ankle samples were sectioned 0.5 cm above and below the joint line, separated from the skin quickly, and fixed in 4% paraformaldehyde for 24 h before decalcification in EDTA-2Na. Ankle samples were dehydrated in a graded series of alcohol and embedded in paraffin, and then 3 μM sections of ankle were stained with H&E staining. Observation and capture were performed with an automated upright microscope system (Olympus, Tokyo, Japan).

#### 4.1.6. Measurement of IL-1β in Gout Rats

Blood samples were collected from the abdominal aorta 48 h after MSU injection. The serum was separated for measurement of IL-1β using an ELISA kit (Cloud-clone, Wuhan, China) according to its instructions. 

#### 4.1.7. Analysis of the NF-κB Signaling Pathway in Synovial Tissues of Gout Rats by Western Blotting

The total proteins were extracted to observe phosphorylation of IκBα and p65 by lysing the synovial tissues with radioimmunoprecipitation assay (RIPA) lysis buffer containing phenylmethanesulfonyl fluoride (Solarbio, Beijing, China) and phosphatase inhibitor cocktail tablets (Roche, Mannheim, Germany). Proteins in the cytoplasm and nucleus of the synovial tissues were separated by a nuclear-cytosol extraction kit (Applygen, Beijing, China) and further purified with an enhanced RIPA (Applygen, Beijing, China) containing inhibitors of phosphorylation for detection of nuclear translocation of p65. The BCA method (Solarbio, Beijing, China) was performed to determine the amount of proteins. The total proteins were mixed with 4× sodium dodecyl sulfate (SDS) loading buffers (Solarbio, Beijing, China) and heated in boiling water for 10 min. Then, 30 μg of each treated sample was separated on 10% SDS-polyacrylamide gels. Proteins were transferred electrophoretically onto a polyvinylidene difluoride (PVDF) membrane (Millipore, Darmstadt, Germany) at 300 mA for 1.5 h. The membrane was blocked with Tris-buffered saline containing 0.1% Tween-20 (TBST) and 5% skimmed milk powder for 2 h at room temperature. Then, the membrane of total protein was incubated with primary antibodies (p-IκBα, 1:1000, p65, 1:1000, p-p65, 1:1000, CST, Beverly, MA, USA; GAPDH, 1:1000, Proteintech, Rosemont, IL, USA) at 4 °C overnight. Additionally, the membrane of nuclear protein was incubated with primary antibodies (p65, 1:1000, CST, Beverly, MA, USA; PCNA, 1:3000, Abbkine, Redlands, CA, USA), and the membrane of cytoplasm protein was incubated with primary antibodies (p65, 1:1000, CST, Beverly, MA, USA; GAPDH, 1:1000, Proteintech, Rosemont, IL, USA) at 4 °C overnight. 

The next day, immunoreactive bands were detected using HRP-conjugated goat anti-mouse IgG (1:5000, Proteintech, Rosemont, IL, USA) or HRP-conjugated goat anti-rabbit IgG (1:5000, Proteintech, Rosemont, IL, USA) as the secondary antibody in TBST for 1.5 h at room temperature. The protein blots were visualized using an enhanced chemiluminescence (ECL) reagent (Millipore, Darmstadt, Germany). The density of the bands was analyzed with Image J. 

Protein expression of p-IκBα was quantified as the ratio of the specific band to GAPDH in the total protein extract. The phosphorylation of p65 was normalized to p65 in the total protein extract. The nuclear translocation of p65 was calculated by the specific value of the density of p65 in the cytoplasm and the density of p65 in the nucleus. 

#### 4.1.8. Analysis of the NLRP3 Signaling Pathway in Synovial Tissues of Gout Rats by Immunohistochemistry

Slides of synovial tissues were prepared as described in Section 4.1.5 (”Histopathological analysis”). Paraffin slides were pretreated with complex digestive fluid (Boster, Beijing, China) to unmask the antigen. Then, its endogenous peroxidase activity was blocked in the dark using 3% hydrogen peroxide. Before staining, the slides were incubated with 5% normal goat serum for 25 min. All of these operations were performed at room temperature and the slides were washed before the next step. Subsequently, paraffin sections were incubated with primary antibodies (ASC, 1:2000, USA; NLRP3, 1:100, Abcam, Cambridge, MA, USA; caspase-1, 1:400, Proteintech, Rosemont, IL, USA) at 4 °C overnight. Eight hours later, the slides were incubated for 60 min with a polymer auxiliary agent and subsequently incubated for 30 min with HRP-conjugated goat anti-rabbit IgG (ZSGB Biotechnology, Beijing, China). All of these steps were performed at 37 °C, and the slides were rinsed with PBS thrice for 5 min before the next step. Color development was achieved with a DAB kit (ZSGB Biotechnology, Beijing, China). After counterstaining with hematoxylin, the slides were mounted. 

All measurements were performed with an automated upright microscope system (Olympus BX53), and five images from each section were randomly captured by a high-speed color CCD camera (Olympus DP72CCD). Imagine Pro-Plus6.0 software was used to analyze the pictures. The positive immunostained area in the total area under the image field of each section was calculated with integral optical density (IOD).

### 4.2. Cell Experiments

#### 4.2.1. Isolation, Characterization, and Cultivation of Macrophages

The macrophages were isolated from the rat peritoneal cavity according to previous reports [42] with some modifications. Firstly, Brewer’s thioglycollate broth solution (Sigma, Louis, MO, USA), a sterile eliciting agent, was injected into the peritoneal cavity of specific pathogen-free rats. Three days later, the rats were euthanized by rapid cervical dislocation. The abdomen of each rat was soaked with 75% alcohol and an incision was made along the midline with sterile scissors to expose the intact peritoneal wall. Cold harvest medium (20 mL) was intraperitoneally injected carefully so as to avoid puncturing the intestine. Fluid from the peritoneum was aspirated with syringes after kneading the abdomen adequately and then the peritoneal fluid was dispensed into a 50 mL conical polypropylene centrifuge tube on ice. The peritoneal exudate cells were centrifuged at 800 rpm and 4 °C for 10 min, the supernatant was discarded, and the sediment was resuspended with lysis buffer of red blood cells for 15 min. Repeated centrifugation was used to clean the cells, and the cell pellets were resuspended with DMEM/F12. A total of 10^6^ cells were added to six-well culture plates and cultured for 2–4 h at 37 °C; then, non-adherent cells were removed by washing with warm PBS. Isolated macrophages were cultured in complete medium consisting of DMEM/F12 (Corning, NY, USA) with 10% new bovine serum (Sijiqing, Hangzhou, China) and 1% penicillin streptomycin solution (Corning, USA) at 37 °C under the conditions of 5% CO_2_ and 95% O_2_.

The phagocytic function of isolated macrophages was observed by phagocytizing chicken red blood cells. Macrophages (10^6^ total nucleated cells/mL, 1 mL) and chicken red blood cells (1%, 0.5 mL) to make cell slides. Giemsa staining was performed after the cultured cells were fixed.

The characterization of isolated macrophages was carried out by immunocytochemistry. Briefly, cell slides were fixed with 4% paraformaldehyde, incubated with 3% Triton, and blocked with 3% H_2_O_2_. Then, then slides were incubated with CD68 polyclonal antibody (1:200, Bioss, Beijing, China) at 37 °C. Two hours later, the slides were incubated for 30 min with HRP-conjugated goat anti-rabbit IgG (ZSGB Biotechnology, Beijing, China), and color development was achieved with a DAB kit (ZSGB Biotechnology, Beijing, China). Pictures were captured by a high-speed color CCD camera (Olympus DP72CCD).

#### 4.2.2. Cell Viability Assay by CCK8

Isolated macrophages (1.0 × 10^4^ cells/mL) were inoculated in 96-well plates (200 μL/well). PBS was added to the peripheral wells (200 μL/well), and the isolated macrophages were cultured for 24 h. Then, the culture medium was replaced with new culture medium containing various concentrations of lyophilized powder of chicory extract and commercial chicoric acid (Pionner, Wuhan, China) for 24 h incubation. CCK8 solution was added into each well according to the instructions for 4 h at 37 °C. The absorbance of each well was detected at 490 nm wavelength in an enzyme microplate reader (Termo Scientifc, Waltham, MA, USA).

#### 4.2.3. Measurement of IL-1β in Isolated Macrophages

Macrophages (3 × 10^5^ cells/mL) were inoculated in a six-well plate and incubated in complete medium containing MSU, lyophilized powder of chicory extract, and chicoric acid for 24 h. After 24 h, the cultured cell supernatant of each well was collected to measure IL-1β production according to the ELISA kit (Cloud-clone, Wuhan, China).

#### 4.2.4. Analysis of the NF-κB and NLRP3 Signaling Pathways in Isolated Macrophages by Western Blotting

Interventions of drugs in isolated macrophages were the same as described above. The cells were collected and lysed by RIPA containing phenylmethanesulfonyl fluoride and phosphatase inhibitor cocktail tablets. The total protein was extracted and denatured by heating in a water bath for 10 min and separated by 10% SDS-PAGE. The subsequent steps, including the blocking procedure, the incubation of antibodies, protein blot visualization, and quantification, were the same as described for the animal experiment.

### 4.3. Statistical Analysis

The results are expressed as the mean ± standard deviation (SD). The statistical analysis was performed using analysis of variance (ANOVA) followed by Dunnett’s multiple comparisons tests to determine the levels of significance using SPSS20.0 software. A *p*-value of <0.05 was considered statistically significant.

## 5. Conclusions

This is the first study to find that chicory extract suppressed ankle edema and gouty inflammation in experimental rats induced with MSU crystals. Biochemical experiments, in vivo and in vitro, demonstrated that chicory extract and chicoric acid can have the antigout effect by inhibiting activation of both the NF-κB and NLRP3 signaling pathways. This work may contribute to the discovery of a new antigout agent which can address both symptoms and root causes of gout, utilize natural medicine resources, and ease the economic burden of gout patients. 

## Figures and Tables

**Figure 1 ijms-20-04921-f001:**
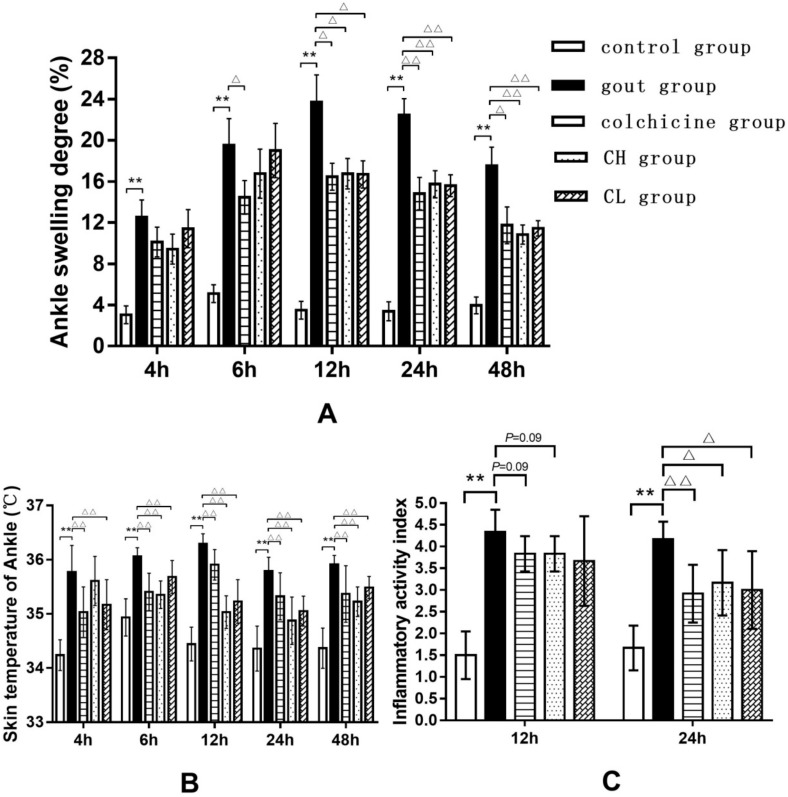
Effect of chicory on ankle edema and inflammatory activity index in MSU-induced gout rats: (**A**) Ankle swelling degree, (**B**) skin temperature of ankle, and (**C**) inflammatory activity index. Data are expressed as mean ± SD for 10 rats in each group. ^**^
*p* < 0.01 vs. control group. ^△^
*p* < 0.05, ^△△^
*p* < 0.01 vs. gout group. Abbreviations: CH, high-dosage chicory treated group; CL, low-dosage chicory treated group.

**Figure 2 ijms-20-04921-f002:**
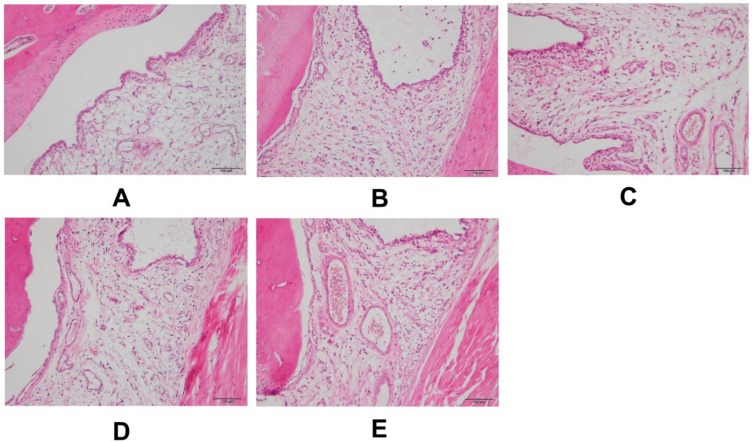
Representative H&E stained slices of synovial tissues in ankle specimens 48 h after intra-articular injection of MSU crystals: (**A**) Control group, (**B**) gout group, (**C**) colchicine group, (**D**) CH group, and (**E**) CL group. 200× magnification.

**Figure 3 ijms-20-04921-f003:**
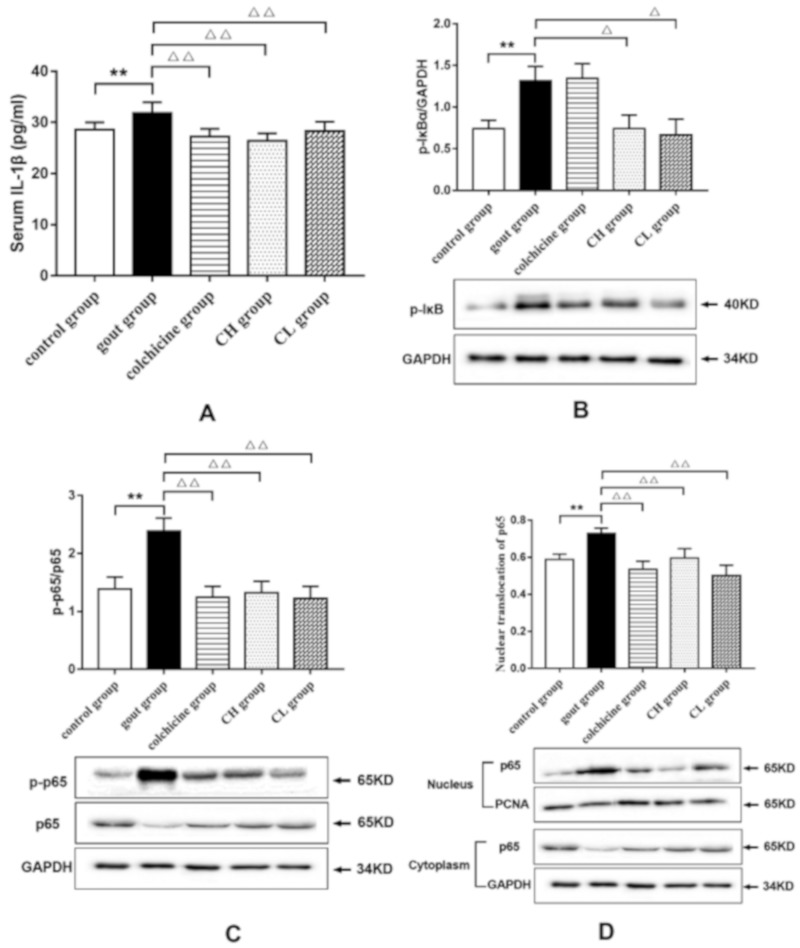
Effect of chicory extract on serum IL-1β and expressions of key targets of the NF-κB signaling pathway in ankle synovial tissues of MSU-induced gout rats: (**A**) Levels of serum IL-1β, (**B**) protein expression levels of p-IκB, (**C**) protein expression levels of p-p65, and (**D**) nuclear translocation of p65. Data are expressed as mean ± SD for six rats in each group. ^**^*p* < 0.01 vs. control group. ^△^
*p* < 0.05, ^△△^
*p* < 0.01 vs. gout group.

**Figure 4 ijms-20-04921-f004:**
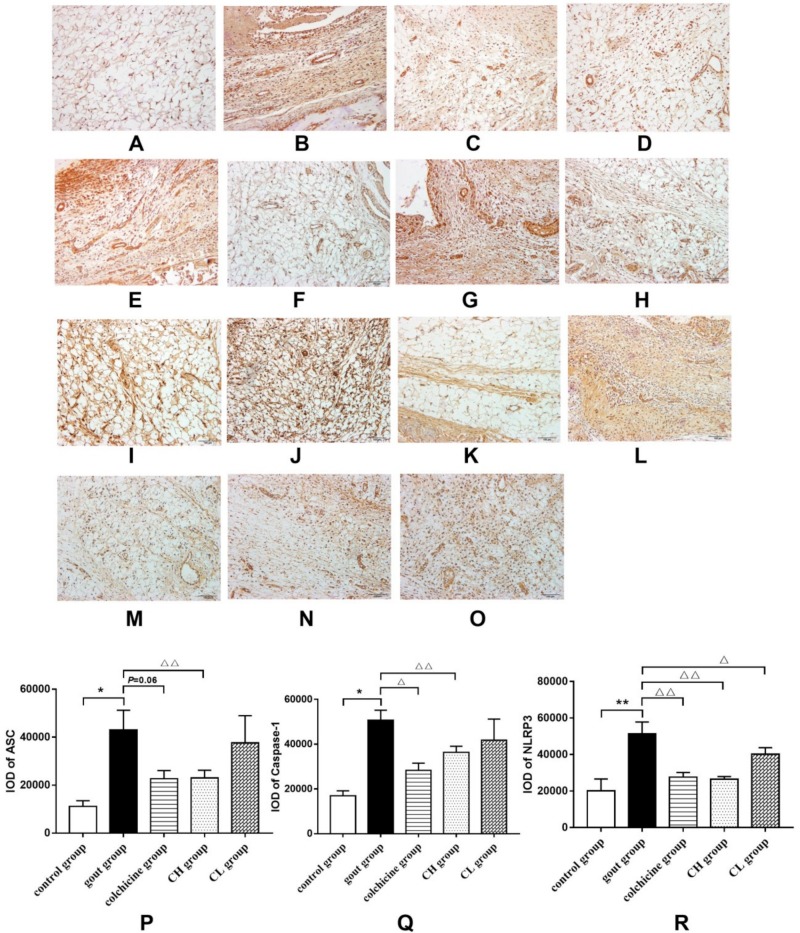
Effect of chicory extract on expressions of inflammasome components in ankle synovial tissues of MSU-induced gout rats: (**A**–**E**) ASC IHC stain, (**F**–**J**) caspase-1 IHC stain, (**K**–**O**) NLRP3 IHC stain, (**P**) protein expression levels of ASC, (**Q**) protein expression levels of caspase-1, and (**R**) protein expression levels of NLPR3. 200× magnification. Data are expressed as mean ± SD for six rats in each group. * *p* < 0.05, ** *p* < 0.01 vs. control group. ^△^
*p* < 0.05, ^△△^
*p* < 0.01 vs. gout group.

**Figure 5 ijms-20-04921-f005:**
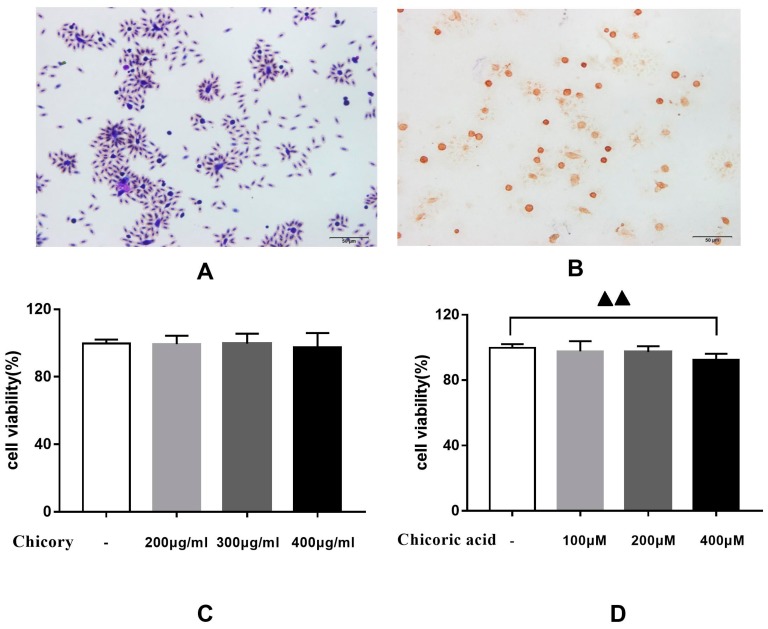
Proliferation ability of macrophages stimulated with various concentrations of chicory extract and chicoric acid: (**A**) Phagocytic function of macrophages, (**B**) CD68 staining in macrophages, (**C**) macrophages stimulated with chicory extract, and (**D**) macrophages stimulated with chicoric acid. Data are expressed as mean ± SD for three repetitions in each group. ^▲▲^
*p* < 0.01 vs. non-treated group.

**Figure 6 ijms-20-04921-f006:**
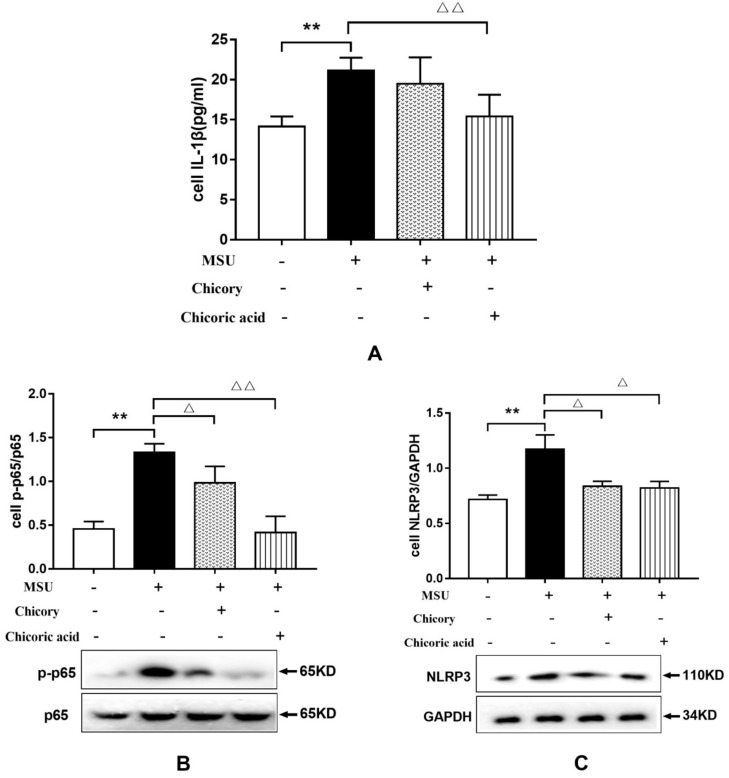
Effect of chicory extract and chicoric acid on IL-1β release and protein expression of p-p65 and NLRP3 in isolated macrophages stimulated with MSU: (**A**) Levels of IL-1β release, (**B**) protein expression levels of p-p65, and (**C**) protein expression levels of NLRP3. Data are expressed as mean ± SD for three repetitions in each group. ^*^*p* < 0.05 vs. non-treated group. ^△^*p* < 0.05, ^△△^*p* < 0.01 vs. MSU-stimulated group.

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
