# Peer review of "Cichorium intybus L. Extract Suppresses Experimental Gout by Inhibiting the NF-κB and NLRP3 Signaling Pathways"

_ijms, 2019, doi:10.3390/ijms20194921_

Round 1

Reviewer 1 Report

The manuscript has the following shortcomings which have to be dealt
with.
Language polishing is required for the manuscript.
Explain why chicoric acid was used only for expts described in Fig 5
and6,but not in all expts.
it appears the keys used for cotrol group and colcichine group look
identical.
Is chicoric acid the only anti-gout principle present in chicory extract?
What is the content of chicoric acid in chicory extract?
Why was uric acid not determined in the study?Explain why in most of
the cases there was no statistically significance between the effects
of high and low doses of chicry extract.It appears that the doses chosen
were not good enough to show a dose-dependent response.
Explain why the apparently statistically significant difference between
the high and low doses in Fig 4P and R was not indicated or spelled out.
A search of the literature discloses the following reports showing the
anti-gout effect of chicory.An effort should be made to mention them
and discuss in the light of these findings as far as possible.
Explain the novelty and contribution of the present investigation in
the light of the published literature.What are the advantages of using
chicory extract or chicoric acid in the treatment of gout over drugs like
allopurinole.g. regarding potency and side effects?

1: Bian M, Lin Z, Wang Y, Zhang B, Li G, Wang H. Bioinformatic and
Metabolomic Analysis Reveal Intervention Effects of Chicory in a
Quail Model of Hyperuricemia. Evid Based Complement Alternat Med.
2018 Dec 3;2018:5730385. doi: 10.1155/2018/5730385. 2: Jin YN, Lin ZJ, Zhang B, Bai YF. Effects of Chicory on Serum Uric Acid, Renal Function, and GLUT9 Expression in Hyperuricaemic Rats with Renal Injury and In Vitro Verification with Cells. Evid Based Complement Alternat Med. 2018 Dec 2;2018:1764212. doi: 10.1155/2018/1764212. eCollection 2018. PubMed PMID: 30622589; PubMed Central PMCID: PMC6304617. 3: Emamiyan MZ, Vaezi G, Tehranipour M, Shahrohkabadi K, Shiravi A. Preventive effects of the aqueous extract of Cichorium intybus L. flower on ethylene glycol-induced renal calculi in rats. Avicenna J Phytomed. 2018 Mar-Apr;8(2):170-178. PubMed PMID: 29632848; PubMed Central 4: Babaei H, Forouzandeh F, Maghsoumi-Norouzabad L, Yousefimanesh HA,
Ravanbakhsh M, Zare Javid A. Effects of Chicory Leaf Extract on Serum
Oxidative Stress Markers, Lipid Profile and Periodontal Status in
Patients With Chronic Periodontitis. J Am Coll Nutr. 2018 Aug;37(6):79-486.
5: Wang Y, Lin ZJ, Nie AZ, Li LY, Zhang B. [Effect of Chinese herb
chicory extract on expression of renal transporter Glut9 in rats
with hyperuricemia].Zhongguo Zhong Yao Za Zhi. 2017 Mar;42(5):958-963. 6: Wang Y, Lin Z, Zhang B, Nie A, Bian M. Cichorium intybus L. promotes intestinal uric acid excretion by modulating ABCG2 in experimental hyperuricemia. Nutr Metab (Lond). 2017 Jun 13;14:38. doi: 10.1186/s12986-017-0190-6. eCollection 2017. PubMed PMID: 28630638; PubMed Central PMCID: PMC5470204. 7: Pourfarjam Y, Rezagholizadeh L, Nowrouzi A, Meysamie A, Ghaseminejad S, Ziamajidi N, Norouzi D. Effect of Cichorium intybus L. seed extract on renal parameters in experimentally induced early and late diabetes type 2 in rats. Ren Fail. 2017 Nov;39(1):211-221. doi: 10.1080/0886022X.2016.1256317. Epub 2016 Nov 16. PubMed PMID: 27846769; PubMed Central PMCID: PMC6014526. 8: Wang XJ, Lin ZJ, Zhang B, Zhu CS, Niu HJ, Zhou Y, Nie AZ, Wang Y. [Molecular docking analysis of xanthine oxidase inhibition by constituents of cichory]. Zhongguo Zhong Yao Za Zhi. 2015 Oct;40(19):3818-25. Chinese. PubMed PMID: 26975108. 9: Zhu CS, Zhang B, Lin ZJ, Wang XJ, Zhou Y, Sun XX, Xiao ML. Relationship between High-Performance Liquid Chromatography Fingerprints and Uric Acid-Lowering Activities of Cichorium intybus L. Molecules. 2015 May 22;20(5):9455-67. doi: 10.3390/molecules20059455. PubMed PMID: 26007193; PubMed Central PMCID: PMC6272355. 10: Lin Z, Zhang B, Liu X, Jin R, Zhu W. Effects of chicory inulin on serum metabolites of uric acid, lipids, glucose, and abdominal fat deposition in quails induced by purine-rich diets. J Med Food. 2014 Nov;17(11):1214-21. doi: 10.1089/jmf.2013.2991. Epub 2014 Oct 14. PubMed PMID: 25314375. 11: Wang Q, Cui J. [A review on pharmic effect of chicory research and development]. Zhongguo Zhong Yao Za Zhi. 2009 Sep;34(17):2269-72. Review. Chinese. PubMed PMID: 19943500.

Reviewer 2 Report

Wang et al., describe “Cichorium intybus L. Extract Suppresses Experimental Gout by Inhibiting the NF-κB and NLRP3 Signaling Pathways”. The authors found that the chicory extract significantly decreased IL-1β release by suppressing the NF-κB and NLRP3 signaling pathways in gout rats. These results suggested chicory extract and chicoric acid may be used as the promising therapeutic agent against gout by inhibiting the NF-κB and NLRP3 signaling pathways.

Comments:

Include a better rationalization for choice of NF-ΚB, NLRP3 and Gout. How this exact NF-ΚB and NLRP3 was chosen was not convincing. The current version of the manuscript lacks a strong discussion. The manuscript should be improved. Scientific writing and careful English grammar editing would also improve this paper. Ex, lines 25: signialing pathways

Author Response

Dear reviewer,

We are very thankful to you for the thorough review and helpful and constructive comments. We have revised our manuscript according to these comments. Our detailed responses to these comments are listed below:

Include a better rationalization for choice of NF-ΚB, NLRP3 and Gout. How this exact NF-ΚB and NLRP3 was chosen was not convincing. The current version of the manuscript lacks a strong discussion. The manuscript should be improved. Scientific writing and careful English grammar editing would also improve this paper.

Response: Thank you for pointing this out. We have revised this section in the revised manuscript to convince discussion and improve readability.

Round 2

Reviewer 1 Report

The manuscript can be accepted in the present form

Reviewer 2 Report

The authors have satisfactorily responded to the comments that the subject matter of this work is acceptable for publication.